# Effects of Environmental Concentrations of Total Phosphorus on the Plankton Community Structure and Function in a Microcosm Study

**DOI:** 10.3390/ijerph19148412

**Published:** 2022-07-09

**Authors:** Xue Bai, Zhendong Jiang, Yuan Fang, Lin Zhu, Jianfeng Feng

**Affiliations:** Key Laboratory of Pollution Process and Environmental Criteria of Ministry of Education and Tianjin Key Laboratory of Environmental Remediation and Pollution Control, College of Environmental Science and Engineering, Nankai University, Tianjin 300071, China; snowyaonuli@163.com (X.B.); 2120200547@mail.nankai.edu.cn (Z.J.); 2120200531@mail.nankai.edu.cn (Y.F.); zhulin@nankai.edu.cn (L.Z.)

**Keywords:** eutrophication, community-level effect, environmentally relevant concentration, principal response curve

## Abstract

The excessive nutrients in freshwater have been proven to promote eutrophication and harmful algae blooms, which have become great threats to water quality and human health. To elucidate the responses of the plankton community structure and function to total phosphorus (TP) at environmental concentrations in the freshwater ecosystem, a microcosm study was implemented. The results showed that plankton communities were significantly affected by the TP concentration ≥ 0.1 mg/L treatments. In terms of community structure, TP promoted the growth of Cyanophyta. This effect was transmitted to the zooplankton community, resulting in the promotion of Cladocera growth from day 42. The community diversities of phytoplankton and zooplankton had been continuously inhibited by TP. The principal response curve (PRC) analysis results demonstrated that the species composition of phytoplankton and zooplankton community in TP enrichment treatments significantly (*p* ≤ 0.05) deviated from the control. For community function, TP resulted in the decline in phytoplankton photosynthesis. The chlorophyll fluorescence parameters were significantly inhibited when TP concentration reached 0.4 mg/L. In TP ≥ 0.1 mg/L treatments, the reductions in total phytoplankton abundances led to a continuous decrease in pH. This study can directly prove that the plankton community changes significantly when TP concentrations are greater than 0.1 mg/L and can help managers to establish specific nutrient management strategies for surface water.

## 1. Introduction

Human activities have changed the contents of nitrogen (N) and phosphorus (P) in aquatic ecosystems, leading to water eutrophication, which is one of the most notable and greatest harms and is expected to continue to increase in the coming decades [1,2,3]. Excessive nutrients cause the development of harmful algae, which can contaminate waterways and inlets, disrupt food web structure, contribute to hypoxia in the freshwater ecosystem, and potentially produce secondary metabolites, which are detrimental to zooplankton, fish, and humans [4,5]. Therefore, it is very important to clarify the impacts of nutrients on aquatic ecosystems. In order to improve the ecological reality, the assessment of the environmental effects of nutrients should be carried out at the community and even ecosystem level. Model ecosystems (i.e., microcosms and mesocosms), which take into account species interactions and identify the latent interactions among biotic community structure and ecosystem function [6], can be used to evaluate the long-term impacts of nutrients in the indoor experiment by collecting parts of natural ecosystems [7,8].

Nutrients have been reported to cause changes in water chemical parameters. The addition of N and P has been reported to increase the electrical conductivity (EC) in water [4], which may be contributed by the elevated concentrations of NO_3_^−^, NH_4_^+^, and PO_3_^−^. When algal blooms occur, especially in the late stage, senescent algae become an important carbon source of microbial food, which can release a large amount of carbon, resulting in the increase in dissolved organic carbon (DOC) concentration in water [9]. The study conducted by Bao et al. has also found that the DOC concentrations are the highest and lowest during cyanobacteria outbreak and incubation, respectively [10]. The responses of phytoplankton community structure to the exogenous nutrients have received much attention in field investigations and bioassay studies [11,12,13]. Due to the species-specific nutrient metabolism and interspecific interaction, increasing the nutrient paradigm can exacerbate competition and succession of the phytoplankton community and can decrease biological diversity, because more tolerant taxa can replace sensitive taxa to be the dominant group [1,14]. One of the most concerned and highly visible symptoms caused by excessive nutrients is the cyanobacterial harmful algal blooms (CyanoHABs), especially for the most common toxin-producing cyanobacteria, including *Microcystis*, *Anabaena*, *Oscillatoria*, etc. [1,15,16,17]. In freshwater ecosystems, excessive N loading could directly and seasonally increase phytoplankton biomass via alleviation of temporal N deficiency, and this conclusion has been confirmed by a number of bioassays and mesocosm experiments [18,19,20]. However, when P concentration is sufficient but N availability is low, N_2_-fixing cyanobacteria bloom in the freshwater ecosystem. In turn, N_2_ fixation by cyanobacteria can help to alleviate N shortages and, hence, maintain a lake in a P-limited condition [21]. Moreover, previous studies have emphasized that the relationships between nutrients and phytoplankton responses in aquatic ecosystems are often nonlinear [11,13]. For example, the study conducted by Xu et al. showed that when the concentration of P is about 0.20 mg/L, the growth rate of cyanobacteria bloom reached the upper limit [5]. It is possible that the impacts of nutrients on the phytoplankton community structure may propagate to zooplankton (bottom-up regulation) [15]. In turn, zooplankton grazing (top-down regulation) might theoretically regulate the effects of nutrient loading on phytoplankton [22]. However, only a few studies have explored the ecological effects of nutrients at the community-level containing the phytoplankton–zooplankton food web at present. The way in which nutrients change the plankton community structure through the classic “trophic cascade” relationships (bottom-up and top-down regulation) has not been clearly concluded [23,24]. The study conducted by Gusha et al. [25] reported that the top-down grazing by zooplankton could not compensate the bottom-up effects of nutrients on phytoplankton. In addition, whether the existence of consumers will change the impact of resource availability on primary producer biodiversity is still controversial [26]. Therefore, our knowledge of the ecological effects of nutrients on plankton community structure is still very limited, and more research needs to be undertaken.

Pollutants can affect community function by altering community structure [27,28]. In the assessment of ecological effects of pollutants, the response of community function is as important as that of community structure [29]. Chlorophyll fluorescence parameters and pH are in vivo probes and indirect physicochemical indicators to characterize the photosynthesis of phytoplankton community, respectively, and can be used to evaluate the impact of pollutants on community function [29]. Species capable of forming harmful algal blooms can reduce the photosynthetic capacity of coexisting algae by releasing allelochemicals after 10 d of exposure [30]. In addition, a large amount of N and P can also promote the photosynthesis of algae, which may prolong the duration of algal bloom and cause harm to the aquatic ecosystem [31,32]. The pH can be used as an indirect physicochemical proxy of photosynthesis, because the algae blooms may consume CO_2_ faster through photosynthesis than what can be replaced by diffusion and ecosystem respiration. The above phenomenon results in the carbonate equilibrium towards HCO_3_^−^ and CO_3_^2−^, eventually leading to an increase in pH [33]. The study conducted by Ma et al. has demonstrated that N and P can result in the increase in pH in water [4]. In turn, the pH of sediment-water interface can mediate the upward transport of P from sediments to the water column of shallow lakes under static conditions [20]. However, the long-term performance of community function along the nutrient gradient at the community level with complex interspecific relationships is still not understood.

Historically, P has been considered as the primary limiting nutrient for the development of phytoplankton in freshwater ecosystems [34]. Meanwhile, according to the Report on the State of the Ecology and Environment in China 2020, TP is the main pollutant of surface water in China [35]. Therefore, we chose total phosphorus (TP) as the environmental stressor in this study. The overall objective of this study was to analyze the impacts of TP on plankton community under environmental concentrations. We conducted indoor microcosms to assess the effects of TP on the plankton community structure (abundance, diversity, species composition, and Algae Group Index (AGI)) and community functional endpoints (in vivo proxy (chlorophyll fluorescence parameters) and indirect physicochemical proxy (pH) of the community photosynthesis). The TP concentrations of enriched treatments were based on the water quality standard of the “Environmental Quality Standards for Surface Water” of China (GB3838-2002) [36]. The main hypotheses were that the responses of the plankton community to TP will be nonlinear. We expected that TP will promote the growth of cyanobacteria, reduce the diversity of phytoplankton community, and then change the structure of zooplankton community and the photosynthetic activity in the microcosm.

## 2. Materials and Methods

### 2.1. Experimental Design

The present experiment was performed by microcosms in a climate-controlled room maintained at 25.6 ± 0.4 °C under a regime of 12 h light: 12 h darkness. Dissolved oxygen (DO) concentrations were maintained at 7.23 ± 0.08 mg/L during the whole experiment by the air aeration system. First, 5 L of natural water and plankton assemblages collected from the Daqing River (117°19′38″ E, 38°59′49″ N) were filled into a beaker (diameter: 17 cm; height: 27 cm) as a microcosm. The freshwater in the microcosm was maintained at 5 L consistently during the experiment. Water losses caused by sampling and evaporation were replenished with *in suit* freshwater, which was filtered with a 0.45 µm glass fiber filter and was stored in the dark. The freshwater in the microcosm was mixed three times a day to promote nutrient circulation and reduce algal settling.

The experiment lasted 63 days. Microcosms were allowed to adapt for 7 days (−7 to 0 d), then treatments were assigned (0 to 56 d). N and P were added according to the target concentration from day 0. The concentrations of TN and TP were determined by the alkaline potassium persulfate digestion UV spectrophotometric method and ammonium molybdate spectrophotometric method [37], respectively, three times a week to maintain a stable concentration. TN was added as KNO_3_ and NH_4_Cl (NO_3_^−^-N/NH_4_^+^-N weigh ratio of 5:1). The concentration of TN was a fixed level (0.80 mg/L, freshwater background value). Xu et al. [5] have reported that the development of phytoplankton was not N limited, when N concentration was closed to 0.80 mg/L. TP was added as K_2_HPO_4_·3H_2_O. The experiment was carried out with various TP concentrations: 0.04, 0.10, 0.20, 0.30, 0.40 mg/L. The TP concentration of control was based on the background values of freshwater. The TP concentrations of enriched treatments were based on the Grades II to V of GB3838-2002 [36]. There were three replicates in each treatment, and a total of 15 microcosms were set.

### 2.2. Turbidity Measurement

In order to prevent algae floating in the mixing process from affecting the turbidity results, the water samples were sampled before mixing the water in the microcosm. The determination process can be seen in “Water Quality-Determination of turbidity” of China (GB 13200-91) [38]. Changes in turbidity in microcosms under different TP concentrations are provided in Appendix A.

### 2.3. Plankton Sampling and Determination

Glass tubes (length: 10 cm; volume: 30 mL) were used to collect samples from well-mixed microcosms to determine the community composition of phytoplankton and zooplankton. The captured plankton species were identified to the lowest practical taxonomic (species or genus) level by microscopes (DMIL, Leica, Germany; Eclipse Ci, Nikon, Japan). For phytoplankton, 30–50 mL water was sampled directly from the microcosm every week and then preserved with 1.5% Lugol’s solution. In order to obtain the responses of phytoplankton taxa groups to TP, the captured phytoplankton species were divided into phyla [14]. For zooplankton, 200 mL water was sampled from the microcosm every two weeks and was filtered through a 64-μm plankton mesh immediately. The filtered water was returned to the microcosms. The zooplankton samples were preserved with 4 mL formalin (37–40% formaldehyde solution). In order to obtain the responses of zooplankton taxa groups to TP, we classified zooplankton species into Rotifera, Cladocera, and Copepoda [39].

### 2.4. Functional Parameter Measurement

We measured the functional parameters characterizing the photosynthetic activity of the community after lights-on. The chlorophyll fluorescence parameters were recorded by the Water-PAM Chlorophyll Fluorometer (PAM-2000, Heinz Walz GmbH, Effeltrich, Germany). The water in the microcosm was homogeneously mixed. Then, 3 mL of water samples were collected from microcosms and were adapted in the dark for 15 min. The measured chlorophyll fluorescence parameters included *F*_v_/*F*_m_, *Y*(II), and ETR. *F*_v_/*F*_m_ represents the maximum photochemical quantum yield of photosystem II (PSII), which reflects the potential maximum photosynthetic capacity of phytoplankton; *Y*(II) characterizes the actual photosynthetic efficiency of phytoplankton; ETR is the relative electron transfer rate and is closely related to the changes in phytoplankton light conditions [40]. In order to avoid CO_2_ in the air entering the water and affecting the measurement results, the pH was determined by a pH glass electrode (PH400, Alalis Instruments Technology (Shanghai) Co., Ltd., China) before mixing the water in the microcosm.

### 2.5. Data Analysis

The Levene’s test and the Kolmogorov–Smirnov test were conducted to test the homogeneity and normality of datasets, respectively. The differences between TP enrichment treatments and control at the same sampling day, regarding the abundances of phytoplankton and zooplankton taxa groups, community diversity indices (Shannon–Weaver index (H), Pielou’s evenness index (J), and Margalef’s richness index (M)) (for formulas see Inyang and Wang [41]), Algae Group Index (AGI, for formula see del Arco et al. [42]), species abundance, and functional parameters (*F*_v_/*F*_m_, *Y*(II), ETR, and pH), were evaluated through two-sample *t*-tests by SPSS 17.0. The dominant species in phytoplankton and zooplankton communities refer to the species with relative abundance ≥5% and ≥10%, respectively.

The effects of TP on the phytoplankton and zooplankton community composition during the experiment were analyzed by the principal response curve (PRC) method [43], which was performed using the “*vegan*” package in R v.4.1.0. We performed 999 Monte Carlo permutations to test the overall significance of the TP enrichment treatment regime on the variation in the community composition. The Dunnett-Contrasts test was used to evaluate the significant difference between TP-enriched treatments and the control at each sampling day [44]. In the PRC diagram, the *x*-axis indicates the sampling days. The *y*-axis indicates the difference between the treatments of TP enrichment and control (canonical regression coefficients (C_dt_)). The species weight (b_k_) can be interpreted as the affinity of species with the PRCs (only species which absolute b_k_ > 0.5 are displayed). In our study, all abundance data were ln(ax+1) transformed before analysis to maintain a minimum species abundance value of 2, where x is the abundance value (for details see Van den Brink et al. [45]).

## 3. Results

### 3.1. Responses of Plankton Community Structure to TP

#### 3.1.1. Responses of Phytoplankton Community Structure

A total of 93 species were collected in all phytoplankton samples throughout the experiment. The sum of Cyanophyta, Chlorophyta, and Bacillariophyta occupied 99.81% of the total abundances, so the present study mainly clarified the changes in these three dominant taxonomic groups. Total abundances of phytoplankton were declined in TP concentration ≥ 0.10 mg/L treatments (Figure 1A). The abundance and dominant position of Cyanophyta were consistently higher in all TP-enrichment microcosms compared to the controls from day 14 (Figure 1B and Appendix A). Both the abundances of Chlorophyta and Bacillariophyta generally tended to decrease in the treatments of ≥0.10 mg/L TP, and the responses of Chlorophyta were more significant and stable (Figure 1C,D).

Figure 2 shows the changes in phytoplankton community diversity during the exposure period. Compared with control groups, the Shannon–Weaver diversity indices (H) and Margalef richness indices (M) were consistently declined in TP ≥ 0.1 mg/L treatments. The eutrophic metric Algae Group Index (AGI), which is one of the indicators raised in the Water Framework Directive (WFD) [42], was also calculated (Appendix A). The AGI close to 1 indicates the good status of the aquatic ecosystem [46]. In our study, the addition of TP disrupted the health of the water body. Compared to controls, the AGI were increased 80.66%, 58.03%, 80.79%, and 116.88% in TP 0.10 mg/L, TP 0.20 mg/L, TP 0.30 mg/L, and TP 0.40 mg/L, respectively.

The multivariate analysis (PRC) results for the phytoplankton community are presented in Figure 3A and Appendix A. The Monte-Carlo test indicated that the effect of TP on phytoplankton species composition is significant (*p* = 0.001). Of all variances, 36.57% could be attributed to treatments, and 24.76% could be attributed to sampling time. The Dunnett-Contrasts test indicated that the responses of the species composition in TP 0.10 mg/L and TP 0.20 mg/L treatments were not significantly different from the control groups in the middle of the experiment. This may be due to the transient decrease in *Oscillatoria tenuis* and/or *Phormidium corium* (Figure 3B), which contributed most (highest positive species score (b_k_)) to the overall response of the community [43]. Generally, all TP enrichment treatments significantly (*p* ≤ 0.05) deviated from the control during the exposure period. To complete the species composition information, we summarized the dominant species (relative abundance ≥ 5%) that were sensitive to community response (absolute b_k_ > 0.5) (Figure 3B). The TP generally led to the increase in the abundances of species belonging to cyanobacteria (e.g., *Anabaena oscillarioides*, *Microcystis aeruginosa*, *Phormidium corium*, and *Oscillatoria tenuis*). The abundances of other species belonging to Chlorophyta (e.g., *Ankistrodesmus falcatus*, *Chlamydomonas ovalis*, *Chlorella vulgaris*, *Crucigenia tetrapedia*, and *Schroederia spiralis*) and Bacillariophyta (e.g., *Synedra* sp.1) were declined from TP 0.10 mg/L upward on at least two consecutive sampling dates.

#### 3.1.2. Responses of Zooplankton Community Structure

A total of 35 taxa (26 Rotifera taxa, 4 Cladocera taxa, and 5 Copepoda taxa) were identified from the zooplankton communities taken from the 15 microcosms. The responses of the zooplankton taxa group abundance to TP are shown in Figure 4 and Appendix A. Compared with the control, the total abundances of zooplankton showed slightly upward trends in TP ≥ 0.10 mg/L treatments from day 28. The abundances of Rotifera and Copepoda were not affected by TP in general. Rotifera was dominant in all treatments. Cladocera began to be captured in TP enrichment microcosms from day 28. The high abundance of Cladocera in TP 0.10 mg/L and TP 0.20 mg/L treatments may be responsible for the decrease in the abundances of *Oscillatoria tenuis* and/or *Phormidium corium* (Figure 3) due to the predation. From day 42, the abundances of Cladocera were promoted by TP significantly (*p* ≤ 0.05); however, the developments of *Oscillatoria tenuis* and/or *Phormidium corium* were not effectively controlled (Figure 3).

The community diversity indices were sensitive to TP concentrations and showed downward trends (*p* ≤ 0.05) (Figure 5). During the exposure period from 0 to 42 days, Shannon–Weaver diversity (H) in 0.10, 0.20, 0.30, and 0.40 mg/L treatments were 35.73%, 28.09%, 39.83%, and 37.79%, respectively, lower than that in the control groups. The results for Pielou evenness (J) showed roughly the same trend. Margalef richness (M) in 0.10, 0.20, 0.30, and 0.40 mg/L treatments were strongly inhibited by 40.56%, 32.90%, 48.40%, and 41.64%, respectively, during the entire period of exposure.

The PRC is significant (*p* = 0.001) and 28.44% of the total variance in the zooplankton dataset is explained by treatment and 52.54% by the factor time (Figure 6A and Appendix A). Statistically significant (*p* ≤ 0.001) differences in zooplankton species composition between controls and TP enrichment treatments during the entire period of exposure. At the species-level, there was a generally significant increasing trend in the abundances of *Monostyla pyriformis* and *Alona rectangular* in TP enrichment treatments. In contrast, the abundance of *Pompholyx complanate* showed a significant downward trend (Figure 6B).

### 3.2. Responses of Plankton Community Functional Endpoints to TP

The responses of chlorophyll fluorescence parameters to TP are shown in Figure 7A–C. When TP concentration was up to 0.40 mg/L, *F*_v_/*F*_m_ was 11.69%, generally lower than that in the control groups. *Y*(II) in 0.10, 0.20, 0.30, and 0.40 mg/L treatments were declined 3.41%, 15.27%, 6.84%, and 14.33%, respectively. The experimental results for ETR showed roughly the same trend. The consistently significant effects (*p* ≤ 0.05) of TP on chlorophyll fluorescence parameters mainly occurred in the TP 0.4 mg/L group. The pH showed a decreasing trend with increasing TP concentration, with a maximum reduction of 2.36% in the TP 0.4 mg/L group (Figure 7D).

## 4. Discussion

### 4.1. Plankton Community Structure Succession along the TP Gradients

In our bioassay, except for cyanobacteria, the abundances of phytoplankton and zooplankton taxa groups showed decreasing trends in TP enrichment treatments before day 14. The reason for such effects may be that TP significantly promoted the growth of *Microcystis aeruginosa*, which can produce microcystins (MCs) during its growth and decomposition [47]. MCs may be toxic to aquatic organisms by inducing the elevated ROS [48,49]. Turbidity can characterize the content of particles in water, reflect the abundance of phytoplankton, and is related to the metabolites of aquatic organisms [50]. In our study, TP reduced the total abundance of phytoplankton, resulting in a reduction in turbidity. Compared with the control, the turbidity was continuously and significantly reduced in TP 0.40 mg/L treatment (Appendix A). The effects of nutrients on the phytoplankton community composition varied with taxa groups, especially true for Cyanophyta, Chlorophyta, and Bacillariophyta [14]. Our study found that TP promoted the development of cyanobacteria and led to an increase in its dominant position in phytoplankton communities after day 7 (Figure 1B and Appendix A). Because cyanobacteria with heterocysts (e.g., *Anabaena oscillarioides*) can fix molecular nitrogen, many ecologists have traditionally considered that the enrichment of P can result in the dominance of cyanobacteria in eutrophic and/or hypereutrophic freshwater ecosystems [21,51]. Meanwhile, the increase in filamentous and/or colonial cyanobacteria (e.g., *Oscillatoria tenuis*) in eutrophic freshwater ecosystems may be due to the declined surface area-to-volume and can be compensated by the increased nutrient availability [52]. The competitive advantage of filamentous and/or colonial cyanobacteria in eutrophic freshwater ecosystems also may be attributed to their morphological defense characteristics because the mucus and sheath can prevent them from zooplankton predation, virus and/or bacterial attack [4,52]. The species with aerotopes (e.g., *Microcystis aeruginosa*) among cyanobacteria favor buoyancy and can float to a suitable position where light is ideal for their growth [53,54]. In conclusion, Cyanophyta maintained a dominant position in the competition under high TP conditions (Figure 1B). Both the abundances of Chlorophyta and Bacillariophyta showed declining trends in the treatments of ≥0.10 mg/L TP (Figure 1C,D). The responses of Chlorophyta were more evident and stable. The declines may be due to the balance between the growth rates of phytoplankton and the losses through the “trophic cascade” effects. Chlorophytes have a high demand for nutrients, as reflected in their high growth rates. Hence, chlorophytes can dominate in high nutrient conditions as they are fast-growing and can be the superior competitor compared with the relatively slow-growing cyanobacteria [4]. However, compared with filamentous or colonial cyanobacteria and diatoms with siliceous exoskeletons, the chlorophytes with a palatable size and lack of specialized structure traits can be regarded as the optimal prey for zooplankton [55,56]. For example, *Chlorella* sp. and *Scenedesmus* sp., which were declined in TP enrichment treatments, have been widely reported to be liable to high predation losses due to their suitable size and the high quality as food [55,56,57]. In our study, the total abundances of zooplankton showed a slight increase trend at the high TP level from day 28 (Figure 4A). This indicated that chlorophytes suffered more tremendous predation pressure, which ultimately led to low abundance. For Bacillariophyta, Ganguly et al. [58] have found that the Cyanophyta *Anabaena* sp. shows more remarkable development in specific growth rate than Bacillariophyta *Chaetoceros simplex* after the enrichment of phosphate-phosphorous (PO_4_-P). In addition, the growth of diatoms has an obligate silicon requirement [14]. Nwankwegu et al. [14] conducted a nutrient addition bioassay and found that only Si addition and the combined N, P, and Si addition can stimulate the growth of Bacillariophyta. The above studies have demonstrated that diatoms are at a disadvantaged position in the process of community competition at high TP concentrations. Furthermore, the control mechanisms of bottom-up (TP enrichment) and top-down (consumer community structure) interaction reduced the biological diversity by increasing the relative abundances of cyanobacteria species and decreasing the richness of species (Figure 2 and Figure 3). Excessive nutrients have been regarded as the indicators denoting the state of the phytoplankton community structure and have been widely reported as an essential driver of biodiversity loss, resulting in a simpler and less stable community structure [14,59,60,61]. The conclusions of our research are consistent with these studies.

For the zooplankton community, the abundances of Cladocera showed significant increasing trends at the high TP level after day 28. This is a strategy for adapting to the changes in phytoplankton community structure (bottom-up effect). Compared to Rotifera and Copepoda, Cladocera has a wider spectrum of prey size [55,62]. Meanwhile, cladocerans have a low prey selectivity due to passive suspension feeding behavior [55,62]. Therefore, cladocerans may benefit from freshwater ecosystems with increased cyanobacteria abundance. Meanwhile, top-down control by zooplankton predation may be used to regulate phytoplankton community structure [23,24]. In our research, the role of zooplankton in TP enrichment groups was mainly reflected in reducing the risk of the sustained increase in the total phytoplankton abundance through the predation on chlorophytes. However, the risk of cyanobacteria blooms was not effectively controlled. The relatively weak coupling between cyanobacteria and zooplankton during cyanobacteria growth may be the reason [63]. Furthermore, the addition of TP declined the zooplankton community diversity indirectly by changing the phytoplankton community composition. The study conducted by Groendahl and Fink found that a decline in producer diversity caused by nutrients may lead to a reduction in consumer diversity, giving rise to a feedback loop of extinction among different trophic levels, which may impair the ability of ecosystems to maintain their biodiversity [26]. Our results confirm this conclusion.

### 4.2. Effects of TP on the Functional Endpoints of Plankton Community

Chlorophyll fluorescence can be used to reflect the photosynthetic performance and stress of phytoplankton [64]. In our study, chlorophyll fluorescence parameters showed slight decreases in TP enrichment treatments. PSII activity was suppressed significantly only when the TP concentration reached 0.40 mg/L, i.e., a worst-case scenario of TP pollution in a natural freshwater ecosystem. In fact, at the community level, the responses of phytoplankton chlorophyll fluorescence parameters to TP are affected by various factors. Phytoplankton, especially cyanobacteria, potentially produced allelochemicals and microcystins (MCs), thereby reducing photosynthetic activity of coexisting algae [60,65]. In addition, the observation results are related to the responses of photosynthetic performance of an individual cell to TP and the cumulative changes in phytoplankton community species composition caused by TP, because each algae has its own photosynthetic characteristics [60]. In general, chlorophyll fluorescence parameters were less sensitive to TP than structural endpoints. This may be due to the dominant position of Cyanophyta being increased in TP enrichment treatments. The light capture structure of Cyanophyta is the phycobiliprotein, which can slide between photosystem II (PSII) and photosystem I (PSI). Therefore, it is difficult to compare the maximum values of *F*_v_/*F*_m_ between different species belonging to Cyanophyta [66]. The reduction in pH can be related to the increased content of CO_2_ in water [33,67,68]. The explanation for this phenomenon was that the reduction in total phytoplankton abundance led to the decreased use of CO_2_. In addition, the slight increase in total zooplankton abundance may produce more CO_2_ through respiration.

### 4.3. Implications for the Environmental Management of TP

We performed a microcosm experiment to elucidate the long-term effects of TP on the plankton community and found that TP clearly and consistently affected the community structure and function when concentration reached 0.10 mg/L. Due to luxury consumption, phytoplankton may store enough cellular P for growth [5]. When TP is sufficient, the plankton development may not necessarily be related to the TP concentration in microcosms [5]. Therefore, the responses of the plankton community to TP were nonlinear when TP ≥ 0.10 mg/L. Cao et al. [1] have found that when TP concentration reaches 0.13 mg/L, the abundances of sensitive species in phytoplankton community decrease, and when TP concentration reaches 0.15 mg/L, the abundances of tolerant species are significantly affected. The study conducted by Chen et al. [69] have demonstrated that when TP concentrations reach 0.05–0.10 mg/L, the benthic macroinvertebrate assemblages are significantly changed. The effect concentration of TP on the plankton community obtained in this study is similar to that obtained from the above field investigation results, which, to a certain extent, shows that indoor microcosm technology can be a useful tool to explore the ecological effect of TP on a freshwater ecosystem under environmental concentrations. The test concentrations were set according to the “Environmental Quality Standards for Surface Water” of China (GB3838-2002). This standard plays a very significant role in the management and protection of the freshwater environment in China. China has made a great deal of significant progress in the development of water quality criteria (WQC) since 2008 [70]. The GB3838-2002 needs to be updated to reflect the latest research results in aquatic organism criteria [71]. The obtained results in this study could help managers to establish specific TP management strategies.

## 5. Conclusions

In the present study, we found the TP ≥ 0.1 mg/L treatments promoted the growth of cyanobacteria and reduced the plankton community diversity. Top-down grazing of zooplankton can control the total abundance of phytoplankton but cannot control the development of Cyanophyta. For community function, the photosynthetic activity of the phytoplankton community was inhibited by TP. Our study complements the understanding of the ecological impacts of TP on freshwater ecosystems containing complex plankton interactions. To the best of our knowledge, this is the first study to evaluate the environmental effects of nutrients on the functional parameters of microcosms containing the phytoplankton–zooplankton food web. In this study, we specifically focused on the responses of plankton community structure and function to TP. Future studies require a closer look into the effects of TP on water chemistry characteristics (e.g., the dynamics of essential elements such as sodium and potassium) at more detailed concentrations and try to find out the sudden change points of the plankton community along the nutrient gradients. In addition, in order to be closer to the real environmental scenario and supplement the effects of TP at the ecosystem level, future research should consider the control effect of higher trophic organisms on plankton in the microcosm containing more complex food web interactions (e.g., phytoplankton–zooplankton–fish food web), and find out more valuable indicators to characterize ecosystem functions.

## Figures and Tables

**Figure 1 ijerph-19-08412-f001:**
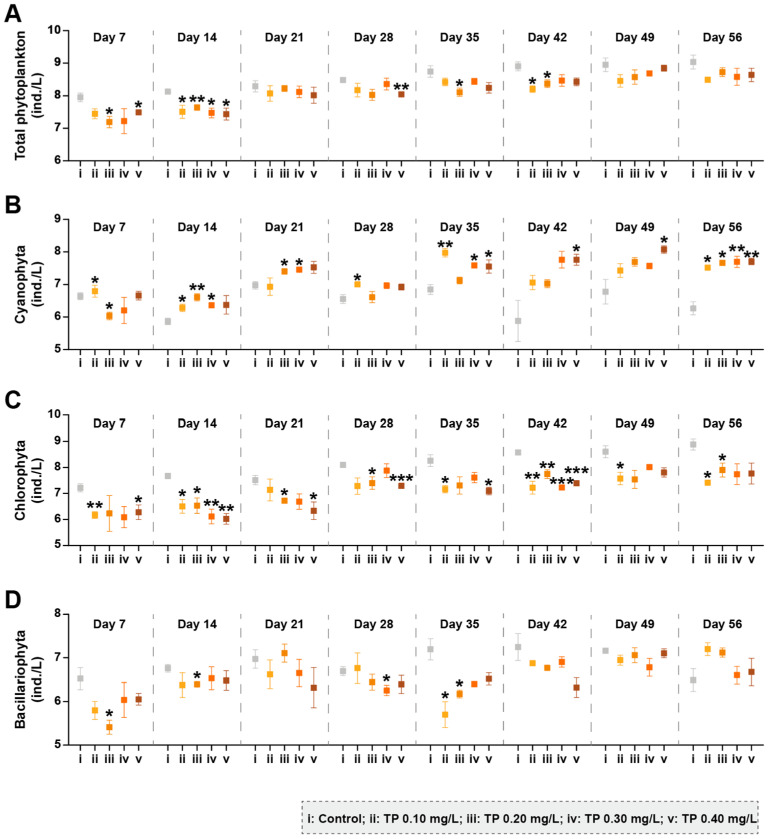
The abundances of phytoplankton taxa groups under different total phosphorus (TP) concentrations. (**A**–**D**) represents the abundance of total phytoplankton, Cyanophyta, Chlorophyta, and Bacillariophyta, respectively. The abundance data were ln(ax+1) transformed. Significant differences are indicated in the figure, *p* ≤ 0.05 is marked as *, *p* ≤ 0.01 is marked as **, and *p* ≤ 0.001 is marked as ***.

**Figure 2 ijerph-19-08412-f002:**
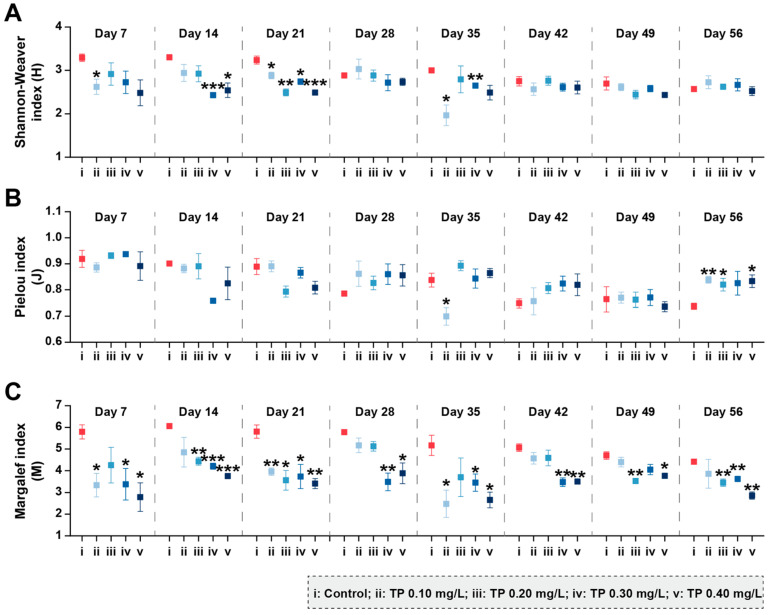
Shannon–Weaver diversity (**A**), Pielou evenness (**B**), and Margalef richness (**C**) of phytoplankton communities under different TP concentrations. Significant differences are indicated in the figure, *p* ≤ 0.05 is marked as *, *p* ≤ 0.01 is marked as **, and *p* ≤ 0.001 is marked as ***.

**Figure 3 ijerph-19-08412-f003:**
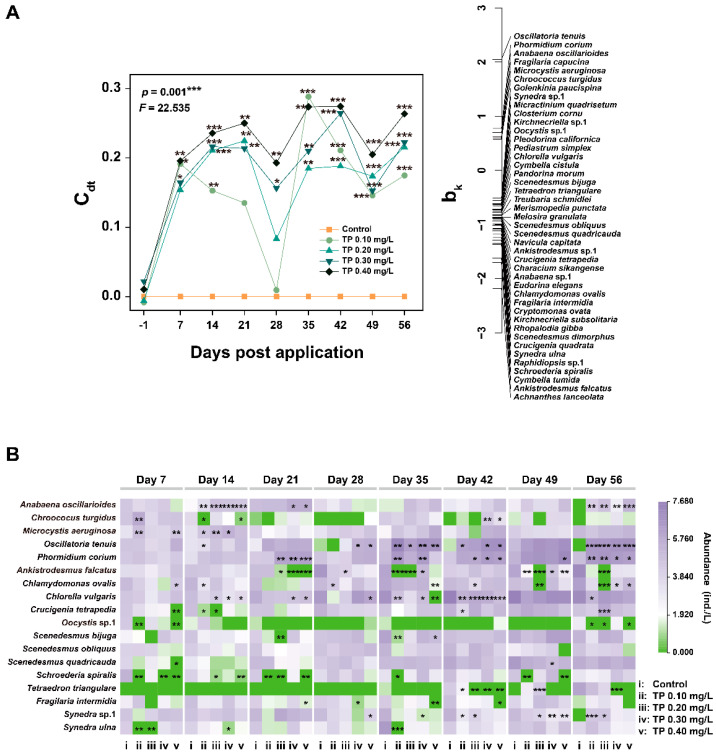
Principal response curve (PRC) diagram (**A**) and the responses of species abundance (**B**) to TP of phytoplankton datasets. The abundance data were ln(ax+1) transformed. Significant differences are indicated in the figure, *p* ≤ 0.05 is marked as *, *p* ≤ 0.01 is marked as **, and *p* ≤ 0.001 is marked as ***.

**Figure 4 ijerph-19-08412-f004:**
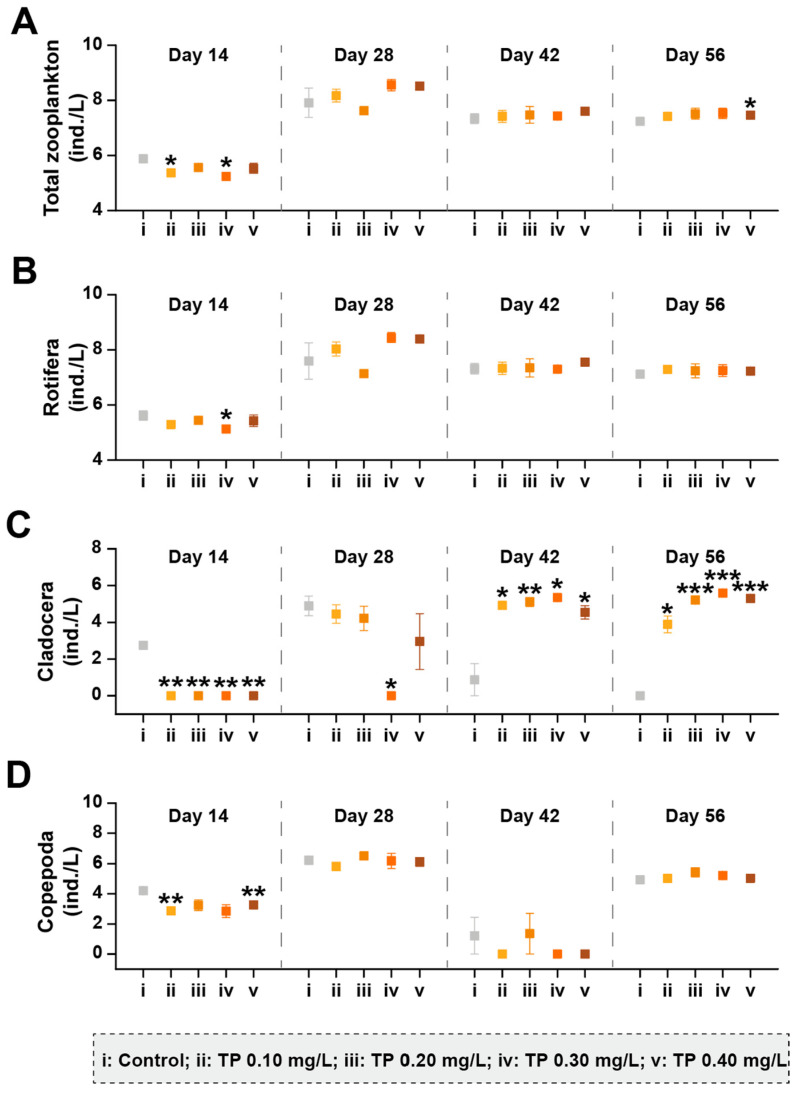
The abundances of zooplankton taxa groups under different TP concentrations. (**A**–**D**) represents the abundance of total zooplankton, Rotifera, Cladocera, and Copepoda, respectively. The abundance data were ln(ax+1) transformed. Significant differences are indicated in the figure, *p* ≤ 0.05 is marked as *, *p* ≤ 0.01 is marked as **, and *p* ≤ 0.001 is marked as ***.

**Figure 5 ijerph-19-08412-f005:**
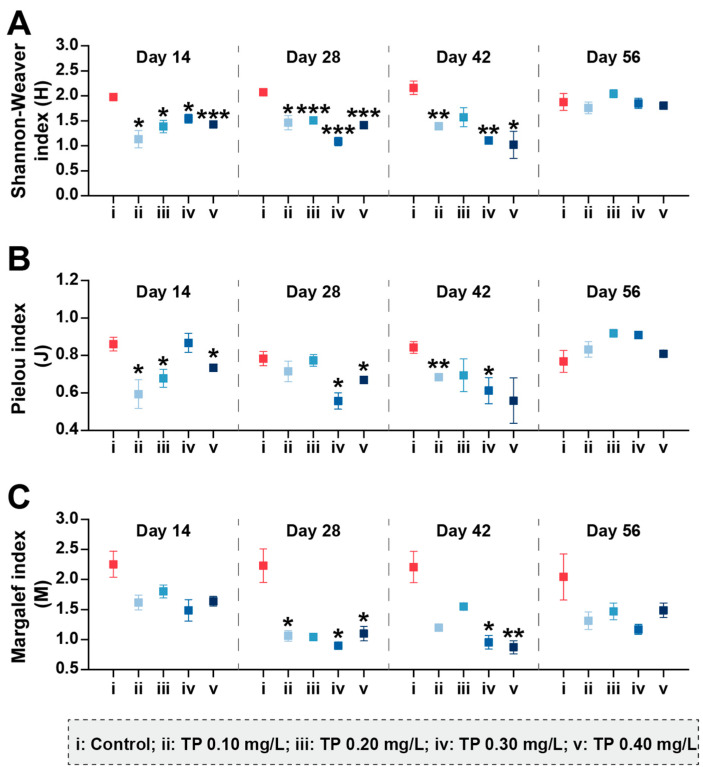
Shannon–Weaver diversity (**A**), Pielou evenness (**B**), and Margalef richness (**C**) of zooplankton communities under different TP concentrations. Significant differences are indicated in the figure, *p* ≤ 0.05 is marked as *, *p* ≤ 0.01 is marked as **, and *p* ≤ 0.001 is marked as ***.

**Figure 6 ijerph-19-08412-f006:**
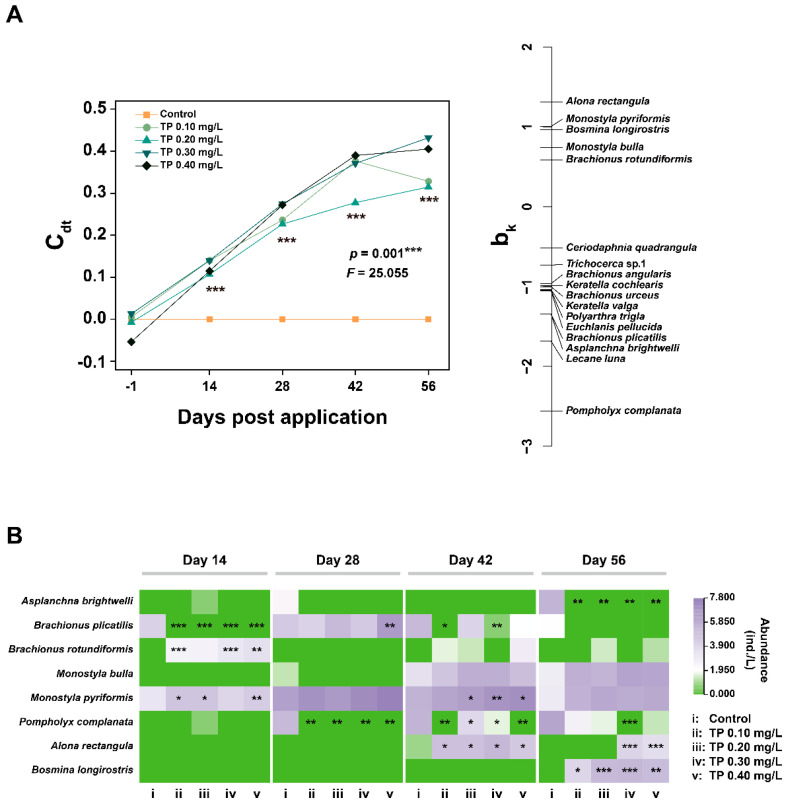
PRC diagram (**A**) and the responses of species abundance (**B**) to TP of zooplankton datasets. The abundance data were ln(ax+1) transformed. Significant differences are indicated in the figure, *p* ≤ 0.05 is marked as *, *p* ≤ 0.01 is marked as **, and *p* ≤ 0.001 is marked as ***.

**Figure 7 ijerph-19-08412-f007:**
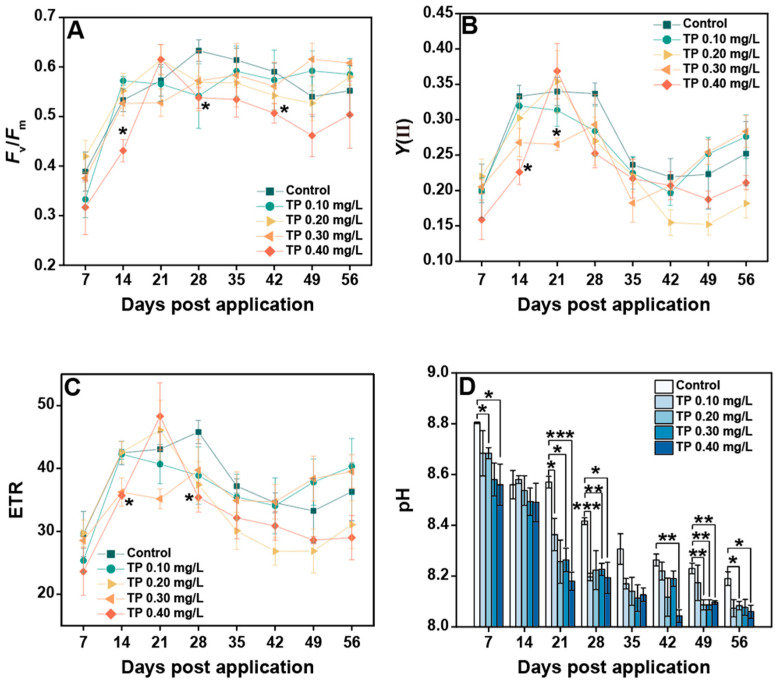
Changes in plankton community functional parameters under different TP treatments. (**A**–**D**) represents *F*_v_/*F*_m_, *Y*(II), ETR, and pH, respectively. Significant differences are indicated in the figure, *p* ≤ 0.05 is marked as *, *p* ≤ 0.01 is marked as **, and *p* ≤ 0.001 is marked as ***.

## Data Availability

The data of this study are available from the corresponding author upon reasonable request.

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
