# Peer review of "Effects of Environmental Concentrations of Total Phosphorus on the Plankton Community Structure and Function in a Microcosm Study"

_ijerph, 2022, doi:10.3390/ijerph19148412_

Round 1

Reviewer 1 Report

General comment: This paper evaluates the ecological effects of TP on the plankton community structure and function at environmental concentrations. Content of manuscript falls adequately within the aims and scope of the Journal and contains interesting data. However, there are several questions in the manuscript which needs to be addressed before accepting the article.

Comment #1: The “introduction” part clarifies the aim of this study, but the scientific hypothesis still needs to be defined.
Comment #2: In the "materials and methods" part, the details of collecting samples characterizing photosynthesis are unclear. For example, whether the sampling is carried out in the light period or in the dark period.
Comment #3: Figure 3A and 6A-change the color of the figure to better distinguish different groups.
Comment #4: Need more fully discussion in the 4.3 part, e.g. highlighting the strengths of the current study.
Comment #5: Some grammatical and editorial errors need to be rectified, e.g. “in suit” (line #106) with italics.

Author Response

General comment: This paper evaluates the ecological effects of TP on the plankton community structure and function at environmental concentrations. Content of manuscript falls adequately within the aims and scope of the Journal and contains interesting data. However, there are several questions in the manuscript which needs to be addressed before accepting the article.

Comment #1: The "introduction" part clarifies the aim of this study, but the scientific hypothesis still needs to be defined.

Response: Thank you for this kind comment. We clarified the research hypotheses in line 114-118 (page 3) as follows: "The main hypotheses were that the responses of the plankton community to TP will be nonlinear. We expected that TP will promote the growth of cyanobacteria, reduce the diversity of phytoplankton community, and then change the structure of zooplankton community and the photosynthetic activity in the microcosm." .

Comment #2: In the "materials and methods" part, the details of collecting samples characterizing photosynthesis are unclear. For example, whether the sampling is carried out in the light period or in the dark period.

Response: We followed this suggestion. To be more clear and in accordance with the reviewer concerns, we have supplemented the details of collecting samples characterizing photosynthesis as follows: "We measured the functional parameters characterizing the photosynthetic activity of the community after lights-on. The chlorophyll fluorescence parameters were recorded by the Water-PAM Chlorophyll Fluorometer (PAM-2000, Heinz Walz GmbH, Germany). The water in the microcosm was homogeneously mixed. Then, 3 mL of water samples were collected from microcosms and were adapted in the dark for 15 min." (line 164-168, page 4); "In order to avoid CO2 in the air entering the water and affecting the measurement results, the pH was determined by a pH glass electrode (PH400, Alalis Instruments Technology (Shanghai) Co., Ltd., China) before mixing the water in the microcosm." (line 173-176 (page 4)).

Comment #3: Figure 3A and 6A-change the color of the figure to better distinguish different groups.

Response: Thank you for the kind advice. We have changed the Figures 3A (page 7) and 6A (page 10).

Comment #4: Need more fully discussion in the 4.3 part, e.g. highlighting the strengths of the current study.

Response: Thanks for this suggestion. We have have added the discussion in line 411 to 419 (page 13) as follows: "Cao et al. [1] have found that when TP concentration reaches 0.13 mg/L, the abundances of sensitive species in phytoplankton community decrease, and when TP concentration reaches 0.15 mg/L, the abundances of tolerant species are significantly affected. The study conducted by Chen et al. [69] have demonstrated that when TP concentrations reach 0.05-0.10 mg/L, the benthic macroinvertebrate assemblages are significantly changed. The effect concentration of TP on plankton community obtained in this study is similar to that obtained from the above field investigation results, which to a certain extent shows that indoor microcosm technology can be a useful tool to explore the ecological effect of TP on freshwater ecosystem under environmental concentration.".

Comment #5: Some grammatical and editorial errors need to be rectified, e.g. "in suit" (line #106) with italics.

Response: Thanks for your kind suggestion. We have carefully scrutinized the manuscript, and made corresponding revisions including some grammatical and editorial errors.

Reviewer 2 Report

The subject of the manuscript is very interesting and important. Finding the mechanism of the reaction of the aquatic ecosystem to the influx of nutrients is very important, especially in the context of industrial development. The authors analysed this problem comprehensively, and it is right. Phytoplankton as well as zooplankton were analysed. The paper is well written. The text is concise, clearly presenting the results and conclusions. The research program is well designed.

My doubt is related to the lack of an analysis of the water chemistry, which also has implications for the organism community.

For a complete picture of the state of the ecosystem, the higher level of the trophic chain should also be analyzed. But I think it is clear to the authors.

line 297 - error in the word "Figure"

Author Response

Reviewer 2

The subject of the manuscript is very interesting and important. Finding the mechanism of the reaction of the aquatic ecosystem to the influx of nutrients is very important, especially in the context of industrial development. The authors analysed this problem comprehensively, and it is right. Phytoplankton as well as zooplankton were analysed. The paper is well written. The text is concise, clearly presenting the results and conclusions. The research program is well designed.

Response: Thank you!

My doubt is related to the lack of an analysis of the water chemistry, which also has implications for the organism community.

Response: We followed this suggestion. In this study, we specifically focused on the responses of plankton community structure and function to TP. As for the responses of water chemistry endpoints to TP, we only measured the changes of turbidity. We have supplemented the determination method of turbidity in line 144-149 (page 3-4), added the research results in the supplementary materials (Figure S1), and discussed the results in line 315-319 (page 11). Nevertheless, we recognize this limitation, so in the conclusion part, we have suggested that further research should focus on relevant analysis (line 436-440, page 14).

For a complete picture of the state of the ecosystem, the higher level of the trophic chain should also be analyzed. But I think it is clear to the authors.

Response: We followed this comment. In the conclusion part, we have emphasized the importance of clarifying the impacts of TP on multiple trophic ecosystems as follows: "In addition, in order to be closer to the real environmental scenario and supplement the effects of TP at the ecosystem level, future research should consider the control effect of higher trophic organisms on plankton in the microcosm containing more complex food web interaction (e.g., phytoplankton-zooplankton-fish food web), and find out more valuable indicators to characterize ecosystem functions." (line 440-445, page 14). In our future work, we will keep the comment in mind.

line 297 - error in the word "Figure"

Response: We feel sorry for our carelessness. In our resubmitted manuscript the error is revised (line 337 (page 12)). 

Reviewer 3 Report

Authors conducted a microcosm bioassay to elucidate responses of plankton community to environmental concentrations of total phosphorus. Obtained data showed significant affect of total phosphorus concentrations on structure of plankton communities, mainly by promoting the growth of Cyanophyta among the phytoplankton, and later transmitting this effect onto zooplankton resulting in growth of Cladocera. Authors also showed that total phosphorus concentrations can inhibit phytoplankton and zooplankton diversity, as well as cause decline in photosynthesis.

The manuscript is clearly written and correctly structured, methods are sufficiently explained, and results are also clearly presented. Obtained results provide useful insight into the ecological impacts of total phosphorus on the plankton interactions in freshwater ecosystems.

General remarks:

I think it would be beneficial if the manuscript was revised by a native speaker, especially the Introduction, as some sentences feal disconnected.

specific remarks:

Introduction

Line 51: I suggest writing “.., including…” instead.

Line 51-52: genus names should be in cursive.

Line 55: Authors seam to alternate between different forms of notation, in some units using superscript (mg P L-1) and slash (mg/L) in others. I think it would be beneficial for manuscript consistency if Authors unify the form of unit notation.

Line 61-62: I’m not sure wat Authors are trying to convey here, please rethink this sentence.

Materials and Methods

Experimental Design

Line 106: Should it be in situ?

Line 112: What kind of “standard method”? Please be more specific.

Plankton Sampling and Determination

I think that section should be rewritten, at first Authors state that organisms were classified into phyla, while later they were identified to lower taxonomic level. Authors also refer to specific species later in the manuscript. This makes the whole section confusing.

Line 129: What formalin solution was used?

Results

Line 185: Please rephrase this sentence.

Discussion

Line 285-286: double “dominance”.

Line: 300-301: “For Chlorophyta,…chlorophytes..” pleas rephrase.

The referee

Author Response

Authors conducted a microcosm bioassay to elucidate responses of plankton community to environmental concentrations of total phosphorus. Obtained data showed significant affect of total phosphorus concentrations on structure of plankton communities, mainly by promoting the growth of Cyanophyta among the phytoplankton, and later transmitting this effect onto zooplankton resulting in growth of Cladocera. Authors also showed that total phosphorus concentrations can inhibit phytoplankton and zooplankton diversity, as well as cause decline in photosynthesis.

The manuscript is clearly written and correctly structured, methods are sufficiently explained, and results are also clearly presented. Obtained results provide useful insight into the ecological impacts of total phosphorus on the plankton interactions in freshwater ecosystems.

Response: Thank you!

General remarks:

I think it would be beneficial if the manuscript was revised by a native speaker, especially the Introduction, as some sentences feal disconnected.

Response: Thank you for your comment. The expressions of the manuscript have been improved with the professional help.

specific remarks:

Introduction

Line 51: I suggest writing ".., including…" instead.

Response: Thanks for your correction. We have corrected it according to your suggestion (line 60, page 2).

Line 51-52: genus names should be in cursive.

Response: We feel sorry for our carelessness. We have revised the error in the resubmitted manuscript (line 60-61, page 2).

Line 55: Authors seam to alternate between different forms of notation, in some units using superscript (mg P L-1) and slash (mg/L) in others. I think it would be beneficial for manuscript consistency if Authors unify the form of unit notation.

Response: We have unified the expression of units in the resubmitted manuscript (line 70, page 2), and we also feel great thanks for pointing out this.

Line 61-62: I’m not sure wat Authors are trying to convey here, please rethink this sentence.

Response: Thank you for your comment. We have corrected it as follows: "The way in which nutrients change the plankton community structure through the classic "trophic cascade" relationships (bottom-up and top-down regulation) has not been clearly concluded [23,24]." (line 76-78, page 2).

Materials and Methods

Experimental Design

Line 106: Should it be in situ?

Response: Thanks for your correction. We have corrected it in line 128 (page 3).

Line 112: What kind of “standard method”? Please be more specific.

Response: We feel sorry that we did not provide enough information about the method. We have supplemented the specific method as follows: "The concentrations of TN and TP were determined by alkaline potassium persulfate digestion UV spectrophotometric method and ammonium molybdate spectrophotometric method [37], respectively, three times a week to maintain a stable concentration." (line 133-135, page 3).

Plankton Sampling and Determination

I think that section should be rewritten, at first Authors state that organisms were classified into phyla, while later they were identified to lower taxonomic level. Authors also refer to specific species later in the manuscript. This makes the whole section confusing.

Response: Sorry for the confusion of plankton sampling and determination. We have rewritten this part in line 151 to 162 (page 4) as follows: "Glass tubes (length: 10 cm; volume: 30 mL) were used to collect samples from well-mixed microcosms to determine the community composition of phytoplankton and zooplankton. The captured plankton species were identified to the lowest practical taxonomic (species or genus) level by microscopes (DMIL, Leica, Germany; Eclipse Ci, Nikon, Japan). For phytoplankton, 30-50 mL water was sampled directly from microcosm every week and then preserved with 1.5% Lugol’s solution. In order to obtain the responses of phytoplankton taxa groups to TP, the captured phytoplankton species were divided into phyla [14]. For zooplankton, 200 mL water was sampled from microcosm every two weeks and was filtered through 64-μm plankton mesh immediately. The filtered water was returned to the microcosms. The zooplankton samples were preserved with 4 mL formalin (37%-40% formaldehyde solution). In order to obtain the responses of zooplankton taxa groups to TP, we classified zooplankton species into Rotifera, Cladocera, and Copepoda [39]. "

Line 129: What formalin solution was used?

Response: We have supplemented the detailed information of formalin solution as follows: "The zooplankton samples were preserved with 4 mL formalin (37%-40% formaldehyde solution)." (line 160-161, page 4).

Results

Line 185: Please rephrase this sentence.

Response: Thanks for your correction. We have corrected it as follows: "Compared with control groups, the Shannon-Weaver diversity indexes (H) and Margalef richness indexes (M) were consistently declined in TP ≥ 0.1 mg/L treatments." (line 219-220, page 6).

Discussion

Line 285-286: double "dominance".

Response: Thank you for pointing this out. We have corrected it as follows: "Because cyanobacteria with heterocysts (e.g., Anabaena oscillarioides) can fix molecular nitrogen, many ecologists have traditionally considered that the enrichment of P can result in the dominance of cyanobacteria in eutrophic and/or hypereutrophic freshwater ecosystems [51]." (line 323-326, page 11).

Line: 300-301: "For Chlorophyta,…chlorophytes.." pleas rephrase.

Response: Thanks for your correction. We have corrected it as follows: "Chlorophytes have a high demand for nutrients as reflected in their high growth rates. Hence, chlorophytes can dominate in high nutrients condition as the fast-growing and can be the superior competitor compared with the relatively slow-growing cyanobacteria [4]. " (line 339-342, page 12).